# Association of plasma acylcarnitines and amino acids with hypertension: A nationwide metabolomics study

Babak Arjmand[1], Hojat Dehghanbanadaki[2], Moein Yoosefi[3], Negar Rezaei[3], Sahar Mohammadi Fateh[3], Robabeh Ghodssi-Ghassemabadi[4], Niloufar Najjar[5], Shaghayegh Hosseinkhani[6], Akram Tayanloo-beik[5], Hossein Adibi[7], Farshad Farzadfar[3], Bagher Larijani[7], Farideh Razi[5]*

1 Cell Therapy and Regenerative Medicine Research Center, Endocrinology and Metabolism Molecular-Cellular Sciences Institute, Tehran, Iran, 2 Diabetes Research Center, Endocrinology and Metabolism Clinical Sciences Institute, Tehran University of Medical Sciences, Tehran, Iran, 3 Non-Communicable Diseases Research Center, Endocrinology and Metabolism Population Sciences Institute, Tehran University of Medical Sciences, Tehran, Iran, 4 Department of Biostatistics, School of Medical Sciences, Tarbiat Modares University, Tehran, Iran, 5 Metabolomics and Genomics Research Center, Endocrinology and Metabolism Molecular-Cellular Sciences Institute, Tehran University of Medical Sciences, Tehran, Iran, 6 Department of Clinical Biochemistry, School of Medicine, Tehran University of Medical Sciences, Tehran, Iran, 7 Endocrinology and Metabolism Research Center, Endocrinology and Metabolism Clinical Sciences Institute, Tehran University of Medical Sciences, Tehran, Iran

☯ These authors contributed equally to this work.
* f-razi@tums.ac.ir

**Data Availability Statement:** All relevant data are within the paper and its Supporting Information files.

## Abstract

### Background

Identification of metabolomics profile in subjects with different blood pressure, including normal blood pressure, elevated blood pressure, stage 1 hypertension, and stage 2 hypertension, would be a promising strategy to understand the pathogenesis of hypertension. Thus, we conducted this study to investigate the association of plasma acylcarnitines and amino acids with hypertension in a large Iranian population.

### Methods

1200 randomly selected subjects from the national survey on the Surveillance of Risk Factors of Non-Communicable Diseases in Iran (STEPs 2016) were divided into four groups based on the ACC/AHA hypertension criteria: normal blood pressure (n = 293), elevated blood pressure (n = 135), stage 1 hypertension (n = 325), and stage 2 hypertension (n = 447). Plasma concentrations of 30 acylcarnitines and 20 amino acids were measured using a targeted approach with flow-injection tandem mass spectrometry. Univariate and multivariate logistic regression analysis was applied to estimate the association between metabolites level and the risk of hypertension. Age, sex, BMI, total cholesterol, triglyceride, HDL cholesterol, fasting plasma glucose, use of oral glucose-lowering drugs, statins, and antihypertensive drugs were adjusted in regression analysis.

**Funding:** The authors received no specific funding for this work.

**Competing interests:** The authors have declared that no competing interests exist.

**Abbreviations:** Acylcarnitines: C0, Free carnitine; C2, Acetylcarnitine; C3, Propionylcarnitine; C3DC, Malonylcarnitine; C4, Butyrylcarnitine; C4DC, Methylmalonyl-/succinylcarn; C4OH, 3-OH-iso-/ Butyrylcarnitine; C5, isovalerylcarnitine; C5:1, Tiglylcarnitine; C5OH, 3-OH-Isovalerylcarnitine; C5DC, Glutarylcarnitine; C6, Hexanoylcarnitine; C8, Octanoylcarnitine; C8:1, Octenoylcarnitine; C10, Decanoylcarnitine; C10:1, Decenoylcarnitine; C12, Dodecanoylcarnitine; C14, Tetradecanoylcarnitine; C14:1, Tetradecenoylcarnitine; C14:2, Tetradecadienoylcarnitine; C14OH, 3-OH-Tetradecanoylcarnitine; C16, Hexadecanoylcarnitine; C16OH, 3-OH-hexadecanoylcarnitine; C16:1OH, 3-OH-Hexadecenoylcarnitine; C16:1, Hexadecenoylcarnitine; C18, Octadecanoylcarnitine; C18:1, Octadecenoylcarnitine; C18OH, 3-OH-Octadecanoylcarnitine; C18:1OH, 3-OH-Octadecenoylcarnitine; C18:2, Octadecadienoylcarnitine; C18:2OH, 3-OH-octadecadienoylcarn.

## Results

Of 50 metabolites, 34 were associated with an increased likelihood of stage 2 hypertension and 5 with a decreased likelihood of stage 2 hypertension. After full adjustment for potential confounders, 5 metabolites were still significant risk markers for stage 2 hypertension including C0 (OR = 0.75; 95%CI: 0.63, 0.90), C12 (OR = 1.18; 95%CI: 1.00, 1.40), C14:1 (OR = 1.20; 95%CI: 1.01, 1.42), C14:2 (OR = 1.19; 95%CI: 1.01, 1.41), and glycine (OR = 0.81; 95%CI: 0.68, 0.96). An index that included glycine and serine also showed significant predictive value for stage 2 hypertension after full adjustment (OR = 0.86; 95%CI: 0.75, 0.98).

## Conclusions

Five metabolites were identified as potentially valuable predictors of stage 2 hypertension.

## Introduction

In recent years, the prevalence of hypertension (HTN) has increased worldwide, especially in low-income countries [1, 2]. In 2011 about 24.1% [3] of the Iranian adult population suffered from hypertension, whereas in 2016, this proportion increased to 29.9% [4], indicating an increasing trend.

HTN is usually diagnosed by measurement of blood pressure by health professionals; However, because of variations in blood pressure, observer bias, and use of nonstandardized devices, HTN may be underdiagnosed [5]. Besides, in many cases, hypertension remains asymptomatic until complications arise. Therefore, using complementary analysis may improve the prediction of hypertension.

Metabolomics has evolved significantly over the years and can now provide us with enormous and valuable data useful for understanding a detailed mechanism in the development of many diseases, including cardiovascular disease. Metabolomics is also emerging as a method for identifying new biomarkers for early diagnosis [6, 7].

It is suggested that disturbances in branched-chain amino acids, urea cycle products (arginine and citrulline), and acylcarnitines may play an important role in cardiovascular disease [8, 9]. Accordingly, this study was designed to investigate the associations between plasma amino acids and acylcarnitines and hypertension in adult Iranians.

## Materials and methods

### Research design

A total of 1200 individuals were randomly selected from a large–scale cross-sectional study of Surveillance of Risk Factors of NCDs in Iran (STEPS 2016). The protocol of STEPS 2016 has been published previously [10]. Through random cluster sampling, 31,050 participants aged $\geq$ 18 years from urban and rural areas of 30 provinces participated in the study, which consisted of collecting demographic data and medical history by completing questionnaires and physical measurements (blood pressure, weight, height, waist circumference, and hip circumference). Ethically, the current study complied with all ethical statements of the Helsinki Declaration and was approved by the Ethics Committee of the Endocrine & Metabolism Research Institute and Tehran University of Medical Sciences (IR.TUMS.EMRI.REC. 1395.00141). The methods and objectives of the study were completely explained to eligible subjects, and written informed consent was obtained before participation.

For accurate measurement of blood pressure (BP), the participant was asked to rest in a sitting position for five minutes. Then, BP was measured three times by qualified personnel using a Lumiscope professional aneroid sphygmomanometer with adult cuff and stethoscope. The average of the last two measurements was counted as the blood pressure reading. BP values were categorized according to the new ACC/AHA hypertension guidelines [11–13] as follows:

Normal: systolic blood pressure (SBP) <120 and diastolic blood pressure (DBP) <80 mmHg

Prehypertension: SBP 120–129 and DBP <80 mmHg

Stage 1 Hypertension: SBP 130–139 or DBP 80–89 mmHg

Stage 2 Hypertension: SBP ≥140 systolic or DBP ≥90 mmHg

## Blood sampling and biochemical measurement

After an overnight fasting, venous blood samples were collected into tubes containing EDTA and sodium fluoride. Plasma specimens were separated. Samples were kept refrigerated or frozen depending on the time of analysis. Moreover, a portion of the whole blood sample was stored for A1c measurement. Biochemical analyses were performed with the Cobas C311 auto-analyzer using commercial kits from Roche Company (Roche Diagnostics, Mannheim, Germany). Two levels of quality control sera were analyzed in each run. The calculated inter-assay precisions (coefficient of variation%) were less than 1.1%, 3%, 3.2%, 1.8%, and 1.7% for glucose, HbA1c, high-density lipoprotein (HDL) cholesterol, triglyceride, and total cholesterol, respectively. In addition, non-HDL cholesterol was calculated by subtracting HDL cholesterol from total cholesterol.

## Metabolomics analysis

A targeted approach was used to measure acylcarnitines and amino acids by flow-injection tandem mass spectrometry (triple quadrupole API 3200 SCIEX with electrospray ionization) coupled with a Thermo Scientific Dionex UltiMate 3000 standard HPLC system using a derivatization method with butanolic-HCL as previously described [14]. Briefly, samples (10 μL) were mixed with the internal standard (IS) and centrifuged at 4˚C. The supernatant liquids were transferred to new tubes and dried with a flow of nitrogen 99.9% at 45˚C. A derivatization solution of acetyl chloride and 1-butanol was added and the vials were incubated at 65˚C for 15 minutes. Samples were dried with a flow of nitrogen 99.9% at 45˚C, and then dissolved in 100 μl of the mobile phase (acetonitrile and water) before injection. Using different dilutions of commercial standards (STD) and based on the response of STD/ IS, calibration curves were generated for each analyte and used to quantify the analyte in each sample. To ensure the reliability of the results, quality control materials were analyzed along with the samples in each run. The mean value of inter-assay precision (coefficient of variation%) was less than 8.7% and 12.3% for amino acids and acylcarnitines, respectively.

## Statistical analysis

After testing data for normality with the Kolmogorov-Smirnov test, quantitative variables were expressed as mean ± SD or median (IQR) and categorical variables were expressed as number (%). Chi-square test and ANOVA test were used to compare the results. Kruskal-Wallis H and Mann-Whitney U tests were used to compare the metabolite concentrations between different groups. To normalize the results of amino acids and acylcarnitines, the natural logarithm of the data was calculated and the value of Z was estimated. The obtained P values were adjusted using Benjamini-Hochberg method. Logistic regression was used to examine the

relationship between metabolites and different stages of hypertension. The correlation between metabolic variables was assessed by Pearson correlation, and a correlation matrix was constructed. Because of the high correlation between metabolites, principal component analysis (PCA) was used to generate independent factors, and the test adequacy was assessed using the Kaiser-Meyer-Olkin (KMO) and Bartlett sphericity tests. The factors based on eigenvalues above 1 were constructed by varimax rotation with maximum probability. Logistic regression analyses were used to examine the relationship between each factor and the different stages of hypertension. All logistic regression analyses were performed in three models, including a univariate model (crude model), model 1 with adjustment for age, sex, and BMI, and model 2 with adjustment for age, sex, BMI, lipid profile (total cholesterol, triglyceride, HDL cholesterol), FPG, intake of oral glucose-lowering drugs, statins, and antihypertensive drugs. We tested the multicollinearity between total cholesterol, triglyceride, and HDL cholesterol and the results showed that the multicollinearity between them was low (S1 Table). The correlation matrix showed a strong linear relationship between FPG and HbA1c (Pearson correlation coefficient = 0.805); because of the multicollinearity between FPG and HbA1c, we selected FPG for adjustment in model 2 because previous evidence in the literature showed that impaired FPG had a 2.13-fold increased risk of HTN, whereas Impaired HbA1c was not independently associated with HTN [15], and FPG also had a stronger correlation with the concentration of various metabolites than HbA1c [16].

## Results

### General characteristics

All randomly selected participants were divided into four groups based on their blood pressure, including 293 subjects with normal blood pressure, 135 subjects with elevated blood pressure, 325 subjects with stage 1 hypertension, and 447 subjects with stage 2 hypertension. The demographic and clinical characteristics of the subjects in each group are shown in Table 1. The results of the post hoc analysis are demonstrated in S2 Table. The sex distribution was the same in each group. Subjects with stage 2 HTN were older than the other groups (p ≤ 0.001). Besides, subjects with stage 1 HTN and also stage 2 HTN were more likely to have higher BMI, waist circumference, FPG, HbA1C, triglyceride, total cholesterol, and non-HDL cholesterol than subjects with normal BP (p ≤ 0.026). The number of subjects taking antihypertensive drugs was 162 (36.2%) in the stage 2 HTN group, 26 (8%) in the stage 1 HTN group, 19 (14.1%) in the elevated BP group, and 16 (5.5%) in the control group.

### The metabolite-metabolite relationships

The correlations between acylcarnitines and amino acids in all participants are depicted in Fig 1. This correlation analysis showed strong positive correlations among medium-chain acylcarnitines including C6, C8, C10, C10:1, and C12; long-chain acylcarnitines including C14, C14:1, C14OH, C16, C16OH, C16:1OH, C16:1, C18, C18:1, and C18:1OH; and a number of amino acids including leucine, methionine, phenylalanine, tyrosine, and valine. The strongest correlations include C8 with C10 (Pearson coefficient = 0.964, p<0.001), C8 with C10:1 (Pearson coefficient = 0.930, p< 0.001), C10 with C10:1 (Pearson coefficient = 0.919, p<0.001), and leucine with valine (Pearson coefficient = 0.910, p<0.001).

### Metabolite profile in different stages of blood pressure

Plasma concentrations of acylcarnitines and amino acids in each study group are shown in Table 2. Of the 30 acylcarnitines and 20 amino acids, the median concentration of alanine and

**Table 1. The demographic and clinical characteristics of participants in different blood pressure categories.**

| Variables | Normal (n = 293) | Elevated BP (n = 135) | Stage 1 HTN (n = 325) | Stage 2 HTN (n = 447) | P value |
|---|---|---|---|---|---|
| Female, n (%) | 157 (53.6) | 74 (54.8) | 161 (49.5) | 235 (52.6) | 0.673 |
| Age (year) | 52.1 (10.4) | 56.2 (12.7) | 53.8 (11.0) | 60.5 (12.1) | **<0.001** |
| BMI (kg/m2) | 26.6 (5.1) | 27.6 (5.2) | 27.8 (5.2) | 28.5 (5.1) | **<0.001** |
| WC (cm) | 91.1 (13.6) | 94.3(14.7) | 94.4 (12.7) | 98.0 (12.2) | **<0.001** |
| HC (cm) | 100.3 (11.8) | 102.2 (11.8) | 102.2 (11.4) | 104.0 (10.4) | **<0.001** |
| SBP (mm Hg) | 109 (7.3) | 123 (2.9) | 127 (5.9) | 153 (16.5) | **<0.001** |
| DBP (mm Hg) | 69 (6.4) | 73 (5.6) | 81 (4.7) | 90 (11.4) | **<0.001** |
| FPG (mg/dL) | 95.1 (18.3) | 101.1 (36.2) | 105.3 (41.8) | 108.6 (43.1) | **<0.001** |
| HbA1C (%) | 5.6 (0.8) | 5.8 (1.0) | 5.9 (1.3) | 6.0 (1.2) | **<0.001** |
| Triglyceride (mg/dL) | 117.4 (66.3) | 135.3 (101.4) | 142.0 (82.4) | 146.1 (122.1) | **0.001** |
| Total cholesterol (mg/dL) | 158.6 (33.5) | 165.8 (33.9) | 171.1 (37.3) | 172.0 (36.4) | **<0.001** |
| HDL cholesterol (mg/dL) | 41.4 (11.6) | 42,9 (12.6) | 40.1 (10.4) | 41.4 (12.0) | 0.106 |
| Non-HDL cholesterol (mg/dL) | 117.2 (33.00) | 122.9 (33.7) | 131.1 (36.3) | 130.5 (36.3) | **<0.001** |
| Smoking, n (%) | 62 (21.2) | 20 (14.8) | 42 (12.9) | 48 (10.7) | **0.001** |
| Medications | | | | | |
| Antihypertensive drugs, n (%) | 16 (5.5) | 19 (14.1) | 26 (8.0) | 162 (36.2) | **<0.001** |
| Oral glucose-lowering drugs, n (%) | 14 (4.8) | 15 (11.1) | 21 (6.5) | 58 (13.0) | **<0.001** |
| Statins, n (%) | 16 (5.5) | 14 (10.4) | 25 (7.7) | 52 (11.6) | **0.025** |

BP: blood pressure; BMI: Body Mass Index; WC: Waist Circumference; HC: Hip Circumference; SBP: Systolic Blood Pressure; DBP: Diastolic Blood Pressure; FPG: Fasting Plasma Glucose; HDL cholesterol: High Density Lipoprotein Cholesterol; Non-HDL cholesterol: Non-High-Density Lipoprotein Cholesterol

valine was higher in the stage 1 HTN group than in the normal BP group ($p \leq 0.025$), and the median concentration of C2, C3, C3DC, C4, C4OH, C4DC, C5, C5OH, C5DC, C6, C8, C8:1, C10, C10:1, C12, C14, C14:1, C14:2, C14OH, C16, C16OH, C16:1OH, C16:1, C18, C18:1, C18OH, C18:1OH, C18:2OH, alanine, leucine, phenylalanine, valine, and proline in the stage 2 HTN group were higher than in the normal BP group ($p \leq 0.047$), and aspartic acid, glycine, and serine had lower concentrations in the stage 2 HTN group than in the normal BP group ($p \leq 0.045$).

Additionally, logistic regression analysis showed that 27 acylcarnitines and 7 amino acids were more likely to increase the odds of stage 2 HTN, whereas 5 amino acids were more likely to decrease the odds of stage 2 HTN. After adjustment for age, sex, and BMI (model 1), 11 metabolites were still associated with the incident of stage 2 HTN. However, when we also adjusted metabolites for lipid profile, FPG, use of oral glucose-lowering drugs, statins, and antihypertensive drugs (model 2), C12 with OR of 1.18 (95%CI: 1.00, 1.40), C14:1 with OR of 1.20 (95%CI: 1.01, 1.42), and C14:2 with OR of 1.19 (95%CI: 1.01, 1.41) were significantly associated with an increased odds of stage 2 HTN, whereas C0 with OR of 0.75 (95%CI: 0.63, 0.90) and glycine with OR of 0.81 (95%CI: 0.68, 0.96) were associated with a decreased odds of stage 2 HTN (S3 Table).

We also performed a factor analysis to reduce the metabolites into few factors that had maximum common variance. This technique resulted in 11 factors, the formula of which is shown in S4 Table. The logistic regression analyses of each factor are depicted in Fig 2. In the crude model, the likelihood of developing stage 2 HTN was higher for each one-unit increase in factor 1 (OR = 1.06; 95%CI: 1.03, 1.09), factor 2 (OR = 1.04; 95%CI: 1.01, 1.06), factor 5 (OR = 1.10; 95%CI: 1.02, 1.18), factor 6 (OR = 1.11; 95%CI: 1.03, 1.20), factor 8 (OR = 1.29; 95%CI: 1.13, 1.48), and factor 9 (OR = 1.22; 95%CI: 1.09, 1.35). In addition, the likelihood of having stage 2 HTN (OR = 0.73; 95%CI: 0.65, 0.82) and stage 1 HTN (OR = 0.85; 95%CI: 0.75,

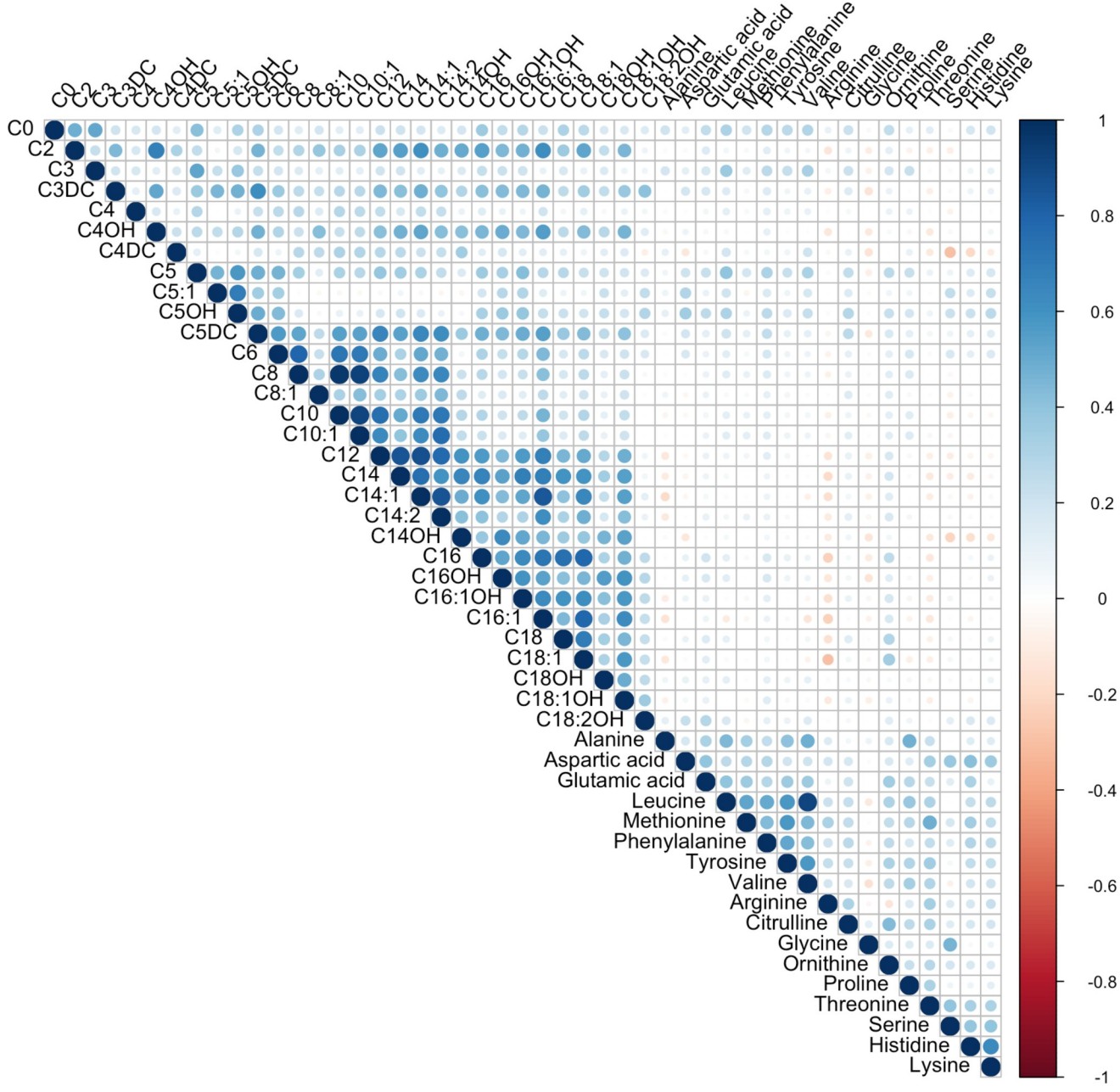

**Fig 1. Heatmap correlation between plasma metabolites in total population.**

0.95) was lower for each unit by which the factor 7 increased. In model 1, the odds of having stage 2 HTN were increased 1.02- and 1.16-fold, respectively, for each unit by which the factor 2 (OR = 1.02; 95%CI: 1.00, 1.05) or factor 8 (OR = 1.16; 95%CI: 1.00, 1.35) was increased. For each unit by which the factor 7 increased, the likelihood of having stage 2 HTN (OR = 0.77; 95%CI: 0.68, 0.88) and stage 1 HTN (OR = 0.88; 95%CI: 0.77, 0.99) decreased. In model 2, the odds of having stage 2 HTN decreased for each unit increase in the factor 7 (OR = 0.85; 95% CI: 0.74, 0.97). Besides, the odds of having elevated BP was lower for each unit increase in factor 6 (OR = 0.86; 95%CI: 0.76, 0.97) and factor 9 (OR = 0.85; 95%CI: 0.73, 0.99).

**Table 2. The plasma concentration of acylcarnitines and amino acids.**

| Metabolites | Normal (n = 293) | Elevated BP (n = 135) | HTN Stage 1 (n = 325) | HTN Stage 2 (n = 447) | P value [*] (FDR) | P value [**] (FDR) | P value [***] (FDR) |
|---|---|---|---|---|---|---|---|
| C0 | 55.378 (47.557–64.450) | 55.132 (45.025–62.727) | 55.712 (48.177–64.300) | 56.114 (47.979–64.640) | 0.616 | 0.966 | 0.647 |
| C2 | 13.567 (11.301–16.894) | 13.323 (11.198–15.328) | 13.869 (11.680–16.468) | 14.453 (11.811–17.258) | 0.616 | 0.729 | **0.044** |
| C3 | 0.771 (0.614–0.946) | 0.708 (0.563–0.987) | 0.840 (0.609–1.148) | 0.867 (0.644–1.090) | 0.792 | 0.107 | **<0.001** |
| C3DC | 0.073 (0.057–0.097) | 0.070 (0.053–0.096) | 0.078 (0.057–0.111) | 0.082 (0.060–0.108) | 0.616 | 0.286 | **0.013** |
| C4 | 0.379 (0.295–0.510) | 0.370 (0.286–0.495) | 0.388 (0.300–0.510) | 0.411 (0.328–0.540) | 0.853 | 0.719 | **0.035** |
| C4OH | 0.051 (0.039–0.064) | 0.052 (0.038–0.063) | 0.052 (0.039–0.069) | 0.053 (0.042–0.072) | 0.899 | 0.495 | **0.016** |
| C4DC | 0.064 (0.048–0.079) | 0.065 (0.051–0.085) | 0.066 (0.051–0.082) | 0.070 (0.055–0.090) | 0.616 | 0.286 | **<0.001** |
| C5 | 0.204 (0.158–0.279) | 0.200 (0.151–0.251) | 0.215 (0.162–0.277) | 0.221 (0.179–0.285) | 0.705 | 0.626 | **0.016** |
| C5:1 | 0.036 (0.027–0.053) | 0.036 (0.027–0.049) | 0.037 (0.028–0.056) | 0.038 (0.030–0.054) | 0.955 | 0.673 | 0.088 |
| C5OH | 0.061 (0.053–0.074) | 0.061 (0.051–0.073) | 0.064 (0.053–0.079) | 0.066 (0.055–0.076) | 0.792 | 0.320 | **0.020** |
| C5DC | 0.289 (0.242–0.372) | 0.285 (0.222–0.364) | 0.307 (0.245–0.386) | 0.322 (0.245–0.411) | 0.625 | 0.626 | **0.014** |
| C6 | 0.161 (0.122–0.228) | 0.145 (0.113–0.221) | 0.170 (0.123–0.247) | 0.178 (0.132–0.258) | 0.616 | 0.669 | **0.017** |
| C8 | 0.243 (0.167–0.352) | 0.229 (0.161–0.337) | 0.254 (0.178–0.375) | 0.269 (0.187–0.394) | 0.625 | 0.495 | **0.016** |
| C8:1 | 0.276 (0.198–0.393) | 0.258 (0.180–0.368) | 0.295 (0.209–0.421) | 0.323 (0.238–0.433) | 0.616 | 0.421 | **<0.001** |
| C10 | 0.312 (0.215–0.446) | 0.283 (0.204–0.444) | 0.326 (0.225–0.516) | 0.357 (0.238–0.538) | 0.705 | 0.421 | **0.006** |
| C10:1 | 0.307 (0.219–0.427) | 0.302 (0.210–0.420) | 0.335 (0.236–0.454) | 0.339 (0.246–0.473) | 0.887 | 0.286 | **0.011** |
| C12 | 0.125 (0.094–0.175) | 0.112 (0.087–0.162) | 0.125 (0.097–0.176) | 0.141 (0.104–0.190) | 0.616 | 0.797 | **0.003** |
| C14 | 0.051 (0.040–0.068) | 0.048 (0.040–0.064) | 0.052 (0.040–0.069) | 0.056 (0.045–0.074) | 0.616 | 0.956 | **<0.001** |
| C14:1 | 0.109 (0.080–0.170) | 0.104 (0.078–0.154) | 0.109 (0.082–0.154) | 0.125 (0.090–0.179) | 0.616 | 0.966 | **0.011** |
| C14:2 | 0.084 (0.064–0.116) | 0.081 (0.061–0.110) | 0.087 (0.065–0.120) | 0.091 (0.068–0.125) | 0.616 | 0.845 | **0.046** |
| C14OH | 0.011 (0.008–0.015) | 0.011 (0.008–0.014) | 0.012 (0.008–0.015) | 0.012 (0.009–0.016) | 0.953 | 0.394 | **<0.001** |
| C16 | 0.169 (0.143–0.207) | 0.171 (0.140–0.203) | 0.170 (0.145–0.207) | 0.185 (0.156–0.226) | 0.899 | 0.966 | **<0.001** |
| C16OH | 0.010 (0.008–0.013) | 0.010 (0.008–0.012) | 0.010 (0.009–0.013) | 0.012 (0.009–0.014) | 0.616 | 0.845 | **<0.001** |
| C16:1OH | 0.016 (0.012–0.021) | 0.015 (0.012–0.019) | 0.016 (0.013–0.021) | 0.018 (0.014–0.023) | 0.616 | 0.939 | **0.003** |
| C16:1 | 0.043 (0.032–0.058) | 0.044 (0.032–0.059) | 0.041 (0.032–0.055) | 0.048 (0.036–0.066) | 0.953 | 0.729 | **<0.001** |
| C18 | 0.062 (0.052–0.076) | 0.060 (0.051–0.072) | 0.060 (0.050–0.077) | 0.064 (0.054–0.083) | 0.616 | 0.865 | **0.033** |
| C18:1 | 0.169 (0.133–0.222) | 0.173 (0.135–0.225) | 0.167 (0.135–0.212) | 0.182 (0.143–0.233) | 0.926 | 0.669 | **0.016** |
| C18OH | 0.008 (0.006–0.010) | 0.008 (0.006–0.010) | 0.008 (0.007–0.010) | 0.009 (0.007–0.011) | 0.853 | 0.286 | **0.003** |
| C18:1OH | 0.011 (0.009–0.015) | 0.011 (0.009–0.014) | 0.012 (0.009–0.014) | 0.013 (0.010–0.016) | 0.792 | 0.966 | **0.009** |
| C18:2OH | 0.027 (0.022–0.035) | 0.027 (0.023–0.036) | 0.028 (0.021–0.036) | 0.028 (0.022–0.037) | 0.705 | 0.669 | **0.020** |
| Alanine | 397.4 (326.8–459.1) | 399.9 (346.0–477.1) | 420.2 (358.2–489.7) | 419.8 (361.6–480.3) | 0.616 | **<0.001** | **<0.001** |
| Aspartic Acid | 12.6 (10.5–15.2) | 11.9 (10.1–14.2) | 12.4 (10.4–14.6) | 12.2 (9.9–14.3) | 0.616 | 0.669 | **0.045** |
| Glutamic Acid | 65.6 (57.6–73.1) | 67.4 (58.3–74.1) | 67.9 (61.5–74.2) | 67.0 (59.3–74.6) | 0.616 | 0.138 | 0.099 |
| Leucine | 119.1 (105.2–134.7) | 126.0 (108.7–138.9) | 126.9 (107.9–144.6) | 124.6 (107.4–141.6) | 0.616 | 0.067 | **0.026** |
| Methionine | 27.7 (24.9–30.8) | 27.0 (24.5–31.9) | 27.6 (24.9–31.6) | 27.3 (24.1–31.9) | 0.853 | 0.929 | 0.429 |
| Phenylalanine | 62.9 (55.9–70.3) | 63.0 (56.7–72.5) | 63.4 (56.4–70.6) | 64.3 (57.4–71.9) | 0.792 | 0.718 | **0.047** |
| Tyrosine | 69.1 (60.1–77.8) | 69.0 (61.2–82.0) | 70.1 (61.0–80.6) | 70.6 (62.4–80.4) | 0.616 | 0.514 | 0.085 |
| Valine | 245.7 (221.0–284.9) | 265.0 (228.1–286.7) | 262.0 (229.5–297.1) | 261.7 (230.3–296.1) | 0.616 | **0.025** | **0.003** |
| Arginine | 68.5 (56.7–81.1) | 68.7 (56.0–84.2) | 70.5 (57.1–84.5) | 67.4 (55.4–79.9) | 0.792 | 0.415 | 0.442 |
| Citrulline | 37.5 (32.0–42.8) | 37.9 (31.0–43.2) | 38.1 (32.2–42.8) | 38.5 (31.7–45.5) | 0.953 | 0.845 | 0.243 |
| Glycine | 263.0 (216.7–331.7) | 262.3 (219.1–344.4) | 242.5 (207.1–308.8) | 240.3 (203.8–296.6) | 0.955 | 0.100 | **<0.001** |
| Ornithine | 86.7 (73.7–100.9) | 86.8 (76.9–103.0) | 88.4 (74.5–102.1) | 89.7 (76.0–107.0) | 0.705 | 0.626 | 0.070 |
| Proline | 232.9 (192.5–285.1) | 240.3 (195.7–283.4) | 251.7 (200.9–300.3) | 243.0 (200.4–304.8) | 0.887 | 0.088 | **0.047** |
| Threonine | 134.2 (116.4–161.0) | 138.2 (117.3–157.1) | 137.4 (116.5–155.2) | 133.8 (113.0–154.2) | 0.926 | 0.966 | 0.104 |

*(Continued)*

**Table 2.** (Continued)

| Metabolites | Normal (n = 293) | Elevated BP (n = 135) | HTN Stage 1 (n = 325) | HTN Stage 2 (n = 447) | P value * (FDR) | P value ** (FDR) | P value *** (FDR) |
|---|---|---|---|---|---|---|---|
| Serine | 105.9 (85.8–125.0) | 100.0 (82.8–119.6) | 98.9 (84.2–117.0) | 95.7 (80.0–111.8) | 0.616 | 0.107 | **<0.001** |
| Histidine | 83.6 (73.5–93.8) | 83.4 (71.2–96.9) | 83.4 (75.4–92.7) | 81.5 (71.6–92.8) | 0.899 | 0.921 | 0.084 |
| Lysine | 180.4 (148.9–202.6) | 179.6 (152.5–204.5) | 180.4 (147.8–204.6) | 174.8 (149.2–208.1) | 0.926 | 0.980 | 0.957 |
| Tryptophan | 68.8 (59.3–77.7) | 71.0 (62.3–82.8) | 70.2 (59.2–81.6) | 68.7 (58.2–79.3) | 0.616 | 0.286 | 0.806 |
| Asparagine | 46.9 (37.8–58.4) | 44.6 (34.6–53.4) | 46.3 (32.1–55.7) | 46.0 (35.4–56.0) | 0.616 | 0.256 | 0.176 |
| Glutamine | 518.1 (436.8–576.9) | 503.4 (453.3–577.3) | 518.1 (433.7–582.5) | 514.3 (430.6–592.0) | 0.853 | 0.966 | 0.934 |

* The pairwise comparison between elevated BP group and normal BP group

** The pairwise comparison between stage 1 HTN group and normal BP group.

*** The pairwise comparison between stage 2 HTN group and normal BP group

## Discussion

In this study, we investigated the plasma profile of a large number of acylcarnitines and amino acids at different stages of blood pressure in a large Iranian population and found that 34 metabolites were associated with an increased odds of stage 2 HTN, and 5 metabolites were associated with a decreased odds of stage 2 HTN. This study also showed that the underlying characteristics of the participants, particularly age, BMI, lipid profile, glycemic profile, use of oral glucose-lowering drugs, statins, and antihypertensive drugs, affected the plasma levels of numerous metabolites in different BP categories, and after full adjustment for these variables, many metabolites lost their predictive value for hypertension, whereas C0, C12, C14:1, C14:2, and glycine remained significant potential biomarkers for predicting stage 2 HTN. This issue not only demonstrated the independent association of these latter metabolites with

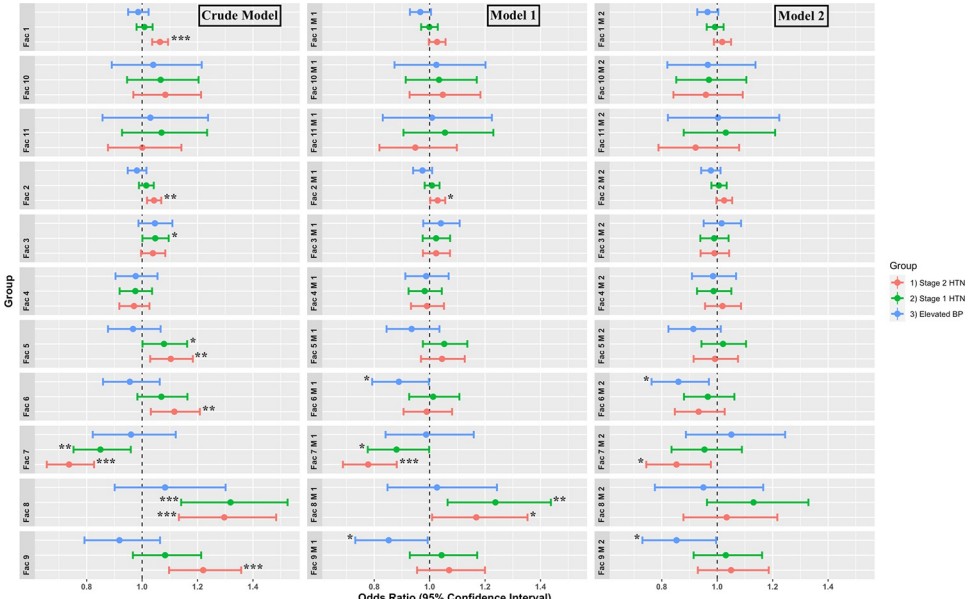

**Fig 2. The likelihood of elevated BP, stage 1 HTN, and stage 2 HTN based on metabolite-derived factors in crude model, model 1 (adjusting for age, sex, and BMI), and model 2 (adjusting for age, sex, BMI, total cholesterol, triglyceride, HDL cholesterol, FPG, oral glucose-lowering drugs use, statins use, and antihypertensive drugs use).**

hypertension but also suggested the potential association of metabolomics profile with age, BMI, lipid profile, and glycemic profile. Besides, we found an index that was predictive of stage 2 HTN independent of underlying risk factors; this index consists of 2 amino acids, including glycine and serine, that was associated with a lower risk of stage 2 HTN.

Acylcarnitines, derived from conjugations of carnitine with acyl-coenzyme A, are important intermediates contributing to the β-oxidation of fatty acids. In this study, the plasma levels of many acylcarnitines were higher in the stage 2 HTN patients. This indicates the inefficiency of β-oxidation in HTN patients, whereas this group requires higher energy consumption to maintain circulation [17]. Besides, we found lower plasma levels of glycine in the stage 2 HTN patients. Glycine is a nonessential amino acid that is involved in the maintenance of normal blood pressure through several mechanisms, including the following by contributing to glutathione synthesis and consequently to the reduction of excess aldehydes and free radicals, by promoting nitric oxide (NO) production, and by participating in the synthesis of elastin and collagen; thus, a deficiency of glycine leads to endothelial dysfunction, impaired availability of NO, and impaired aortic elasticity, resulting in the development of HTN [18–20].

Consistent with our findings, several previous studies have investigated acylcarnitines and amino acids as potential biomarkers for predicting incident HTN [21], ambulatory blood pressure [22], preeclampsia [23], pulmonary hypertension [24], and persistent pulmonary hypertension of the newborn (PPHN) [25]. Acylcarnitine C16:0 and branched-chain amino acids (BCAAs) were found to be strongly associated with incident HTN [21, 26]. Long-chain acylcarnitines were positively associated with ambulatory blood pressure [22] and pulmonary hypertension [24]. C18:0 was identified as a good predictor of early- and late-onset of preeclampsia [23]. Ornithine and tyrosine were also found to decrease the likelihood of PPHN, whereas phenylalanine increased the likelihood of PPHN [25]. The study of four classes of antihypertensive drugs also showed that these agents altered plasma concentrations of acylcarnitines in addition to affecting their target receptors. In this case, amlodipine decreased circulating concentrations of C4OH, C6, C8, C10, C10:1, C12, and C14. Losartan decreased circulating concentrations of C8 and C10. Bisoprolol decreased circulating levels of C4, C8, and C14:1, and hydrochlorothiazide increased circulating levels of C0 [27]. When we included intake of antihypertensive drugs in our fully adjusted model, we found that three metabolites, including C8:1, C10, and C10:1, lost their predictive value for stage 2 HTN. This finding indicated that C8:1, C10, and C10:1 were not associated with a higher likelihood of stage 2 HTN independent of antihypertensive drugs use.

In another Chinese metabolomics study, C8:1 and C10:1 were found to have good diagnostic abilities for pulmonary arterial hypertension (PAH) [28]. These metabolites were decreased in Chinese PAH patients; however, we found higher plasma concentrations of these metabolites in HTN patients compared with normal BP subjects. One explanation for this inconsistency is that previous evidence showed a positive correlation between C8:1 and BMI [29], and the Chinese PAH patients had lower BMI than control subjects, and they did not consider the confounding effect of BMI; however, in our study, C8:1 and C10:1 remained as risk markers for stage 2 HTN when we adjusted for BMI, but this association was dependent of antihypertensive drug use. Besides, another metabolomic analysis of chronic thromboembolic pulmonary hypertension (CTEPH) found lower levels of C10 and higher levels of C10:1, C12, C14:1, and C16 in plasma from CTEPH patients compared with healthy controls [30].

Our crude and adjusted models showed that glycine was negatively associated with the stage 2 HTN. In line with our findings, several previous studies reported lower levels of glycine in HTN [31–33]. Dietary glycine has also been shown to lower hypertension [20]. Nevertheless, dietary serine intake in the Iranian population resulted in a 70% increase in the risk of developing HTN [34]. A combination of blood metabolites, including glycine, C10, C5OH/C8,

phenyalanine/tyrosine, ornithine, and ornithine/citrate, was shown to be highly accurate for predicting essential HTN, with a sensitivity and specificity of 95.16% and 87.50%, respectively [31]. C12 and C14:1 were also elevated in the serum of patients with HTN [17]. These findings were in agreement with ours. Studying the metabolomic characterization of HTN seems to be a promising strategy to understand the mechanisms underlying the development of HTN and ultimately discover new biomarkers for diagnostic, prognostic, preventive, and therapeutic purposes in the future.

### Limitations and strengths

This study had several significant strengths, including the large-scale population-based design, the examination of a large number of acylcarnitines and amino acids profile, and the assessment of the predictive value of each metabolite based on different regression models to eliminate various potential confounders. This study also had some limitations that should be discussed. Because this study was cross-sectional, we could not perform a longitudinal analysis or infer causality. Although antihypertensive drugs affect plasma levels of metabolites and we adjusted for antihypertensive drugs intake accordingly, the type of antihypertensive drugs taken by participants was unclear. In addition, this study was performed on the Iranian population, and we could not generalize our results to other communities with different ethnicity.

### Conclusion

In this study, we demonstrated the pattern of plasma metabolite concentrations in individuals with different blood pressures, including normal blood pressure, elevated blood pressure, stage 1 and stage 2 hypertension, and identified 5 metabolites as potential biomarkers for predicting stage 2 HTN. Our results suggest that high plasma levels of carnitines and various acylcarnitines as well as low plasma levels of glycine may play a crucial role in the development and progression of HTN.

### Supporting information

**S1 Table. The multicollinearity test between total cholesterol, triglyceride, and HDL cholesterol according to Pearson correlation coefficient.**
(DOCX)

**S2 Table. The pairwise comparison of sociodemographic and laboratory parameters between groups.**
(DOCX)

**S3 Table. The logistic regression analysis on the metabolite profile for predicting Elevated BP, stage 1 HTN, and stage 2 HTN.**
(DOCX)

**S4 Table. The formula of 11 factors applying for factor analysis.**
(DOCX)

### Author Contributions

**Conceptualization:** Babak Arjmand, Hojat Dehghanbanadaki, Farshad Farzadfar, Bagher Larijani, Farideh Razi.

**Data curation:** Moein Yoosefi, Negar Rezaei, Sahar Mohammadi Fateh, Niloufar Najjar, Akram Tayanloo-beik, Farshad Farzadfar, Farideh Razi.

**Formal analysis:** Hojat Dehghanbanadaki, Robabeh Ghodssi-Ghassemabadi, Farideh Razi.

**Investigation:** Moein Yoosefi, Negar Rezaei, Sahar Mohammadi Fateh, Niloufar Najjar, Shaghayegh Hosseinkhani.

**Methodology:** Robabeh Ghodssi-Ghassemabadi, Farideh Razi.

**Project administration:** Babak Arjmand, Niloufar Najjar, Akram Tayanloo-beik, Hossein Adibi.

**Supervision:** Farshad Farzadfar, Bagher Larijani, Farideh Razi.

**Validation:** Hojat Dehghanbanadaki.

**Visualization:** Hojat Dehghanbanadaki.

**Writing – original draft:** Hojat Dehghanbanadaki.

**Writing – review & editing:** Babak Arjmand, Hojat Dehghanbanadaki, Negar Rezaei, Robabeh Ghodssi-Ghassemabadi, Shaghayegh Hosseinkhani, Hossein Adibi, Farshad Farzadfar, Farideh Razi.

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
