## [Decision Letter · Decision Letter 0]

24 May 2022

PONE-D-21-34134Association of plasma acylcarnitines and amino acids profile with hypertension: a nationwide metabolomics studyPLOS ONE

Dear Dr. Farideh,

Thank you for submitting your manuscript to PLOS ONE. After careful consideration, we feel that it has merit but does not fully meet PLOS ONE’s publication criteria as it currently stands. Therefore, we invite you to submit a revised version of the manuscript that addresses the points raised during the review process. Please refer to the reviewers' comments attached and revise accordingly.

We look forward to receiving your revised manuscript.

Kind regards,

Jie V Zhao

Section Editor

PLOS ONE

Journal Requirements:

4. Please upload a new copy of Figure 1 as the detail is not clear. Please follow the link for more information: https://blogs.plos.org/plos/2019/06/looking-good-tips-for-creating-your-plos-figures-graphics/" https://blogs.plos.org/plos/2019/06/looking-good-tips-for-creating-your-plos-figures-graphics/

Reviewers' comments:

Reviewer's Responses to Questions

**Comments to the Author**

1. Is the manuscript technically sound, and do the data support the conclusions?

Reviewer #1: Yes

Reviewer #2: Partly

2. Has the statistical analysis been performed appropriately and rigorously? 

Reviewer #1: Yes

Reviewer #2: I Don't Know

3. Have the authors made all data underlying the findings in their manuscript fully available?

Reviewer #1: No

Reviewer #2: No

4. Is the manuscript presented in an intelligible fashion and written in standard English?

Reviewer #1: Yes

Reviewer #2: No

5. Review Comments to the Author

Reviewer #1: This is a cross sectional study examining serum metabolomics profile in subjects with varying levels of hypertension in Iran. 1200 subjects were selected from a national survey. The subjects were stratified according to blood pressure level. It was not clear if any of the subjects were taking medication for blood pressure. Serum was obtained and subjected to metabolomics analysis to measure acylcarnitines and amino acids. Appropriate statistical handling was performed including regression as well as principal component analysis to look for association between serum metabolites and blood pressure levels. After several adjustments for metabolic disease risk factors several acylcarnitine species were associated with higher odds of elevated blood pressure and glycine was associated with lower odds of elevated blood pressure. Subsequently, principal component analysis was applied to look for factors which risk of increased blood pressure. After adjustment a factor containing mostly medium and short chain acylcarnitines was associated with higher risk while factors containing glycine and serine and another containing ornithine and short chain acyl-carnitines were associated with lower risk of elevated blood pressure. The discussion reviews the finding in the context of what is known about metabolomics and hypertension as well as proposing possible biologic mechanisms to explain their findings.

Overall the study was well conducted with a large number of study subjects. The results add to our understanding of serum metabolome in subjects with elevated blood pressure.

One minor comment: I would have liked to know if any of the subjects in the study were on anti-hypertensive medications.

Reviewer #2: Thank you for the opportunity to review the manuscript by Razi et al. In this interesting research article, link between plasma amino acids and acylcarnitines with different blood pressure categories ranging from normal BP to elevated to the different stages of hypertension were investigated. Two metabolite clusters were identified with factor analyses. The first factor consisting of various short- and medium chain acylcarnitines was associated with an increased odds of having stage 2 hypertension, while a second factor consisting of two amino acids, namely glycine and serine was associated with a lower odds of have stage 2 hypertension. I do have the following comments and suggestions:

• Overall, the manuscript would benefit greatly from professional language and technical editing.

• Abstract: Line 27: What is meant with different stages of blood pressure. This should be phrased so that it is clear that you divided your groups based on different blood pressure cut points to classify normal, high and hypertensive stages.

• Abstract: Which criteria did you use for blood pressure classification

• Abstract: Line 36-37: The study design and statistical approach does not allow you to determine cause and effect. Therefore, you should rephrase: “to estimate the effect of metabolites on the risk of hypertension.”

• Abstract: Perhaps indicate the confounders you adjusted for.

• Abstract: Be more specific regarding the indexes (or factors) you identified. It is stated in a much clearer fashion in the conclusion of the article.

• Introduction: Line 53: “which shows a significant growth trend.” Can perhaps be rephrased to indicate that there was an upward trajectory.”

• Introduction: Line 57: “….it might be ignored.” It is not clear what is meant here? What might be ignored?

• Introduction: Line 60-62: The mention of biomarkers not related to metabolomics per se is not applicable here. Perhaps you should rather refer to what was found in previous metabolomics papers?

• Introduction: Line 61-62: “….no parameters have been confirmed for diagnosing and predicting hypertension.” This statement is simply not true as there are many factors previously reported on (including biomarkers) that can be used for the prediction of hypertension. Perhaps I misunderstood you. Please clarify.

• Introduction: Line 66: What is meant with metabolomics is a practical method? It requires expensive equipment and trained staff to perform these analyses?

• Introduction: Line 68: The reference to organic acids is quite broad as there are many organic acids that form part of multiple metabolic pathways?

• Methods: Line 83: Please specify the device used to measure BP

• Methods: For both the biochemical measurements and the metabolomics analysis the coefficient of variation should be reported.

• Methods: What type of QCs did you incorporate in your metabolomics analyses. Was it done in batches? How did you correct for batch effects?

• Methods: Line 101: Reference is made to internal standards. What was used? What concentrations?

• Methods: What is the level of certainty of the identification of your metabolites? Did you compare it with commercially available standards? How did you quantify the levels of your metabolites?

• Methods: Line 121: Adjustments were made for the lipid profile? Did you include all measures as part of the lipid profile? Or only HDL? Be specific.

• Methods: Line 121: How did you decide on FPG to adjust for instead of HbA1c?

• Results: Line 127-128: Instead of indication that each group consisted of almost half females, rather indicate that there was an equal sex distribution.

• Results: Line 131: I am not sure where the (p≤0.026) comes from as all the p-trend values indicated in Table 1 (with the exception of statin use) are 0.001? Did you also compare the groups in post-hoc analyses? If you did, it should be indicated in Table 1.

• Did you adjust for statin use or the use of other medication? Perhaps that should be included as sensitivity analyses?

• What was the rationale for inter-correlating the metabolites (Fig1). As this was not part of your original aim?

• Table 2: It is not clear what exactly you are reporting. Did you compare all groups? Or did you compare all groups with the control group? I think it may be worthwhile to do the latter if you did not. Also, to just indicate that there are differences without indicating if it was higher/lower does not say much?

• Discussion: Line 205 & 246: I think the use of the work “upregulated” is not appropriate here. Rather say that the abundance of acylcarnitines were higher. Normally you would use the term upregulated for an enzyme and in this case if acylcarnitines are accumulated it would suggest that the beta-oxidation enzymes are not functioning optimally?

• Discussion: Line 205-207: It is indicated that beta-oxidation may not be functionally optimally. Can you speculate as to why this is the case?

• Discussion: Line 254: It is indicated that plasma is the best sample choice. It should be considered that urine can also be used and that urine is a lot less invasive to collect.

• Conclusion: Line 261: “….found the unique metabolite profile of plasma” This phrase in unclear. Also the phrase “….different stages of blood pressure” is not clear and should be clarified.

6. PLOS authors have the option to publish the peer review history of their article (what does this mean?). If published, this will include your full peer review and any attached files.

Reviewer #1: **Yes: **Jean-Paul Kovalik

Reviewer #2: **Yes: **Catharina MC Mels

---

## [Author Response · Author response to Decision Letter 0]

16 Jul 2022

Response to Reviewers

Journal Requirements:

Done. 

Done. All relevant data are within the manuscript and its Supporting Information files.

Done.

4. Please upload a new copy of Figure 1 as the detail is not clear. Please follow the link for more information:https://blogs.plos.org/plos/2019/06/looking-good-tips-for-creating-your-plos-figures-graphics/" https://blogs.plos.org/plos/2019/06/looking-good-tips-for-creating-your-plos-figures-graphics/

Done.

Done. Thanks for your precise comment.

Reviewers' comments:

Reviewer #1: This is a cross sectional study examining serum metabolomics profile in subjects with varying levels of hypertension in Iran. 1200 subjects were selected from a national survey. The subjects were stratified according to blood pressure level. It was not clear if any of the subjects were taking medication for blood pressure. Serum was obtained and subjected to metabolomics analysis to measure acylcarnitines and amino acids. Appropriate statistical handling was performed including regression as well as principal component analysis to look for association between serum metabolites and blood pressure levels. After several adjustments for metabolic disease risk factors several acylcarnitine species were associated with higher odds of elevated blood pressure and glycine was associated with lower odds of elevated blood pressure. Subsequently, principal component analysis was applied to look for factors which risk of increased blood pressure. After adjustment a factor containing mostly medium and short chain acylcarnitines was associated with higher risk while factors containing glycine and serine and another containing ornithine and short chain acyl-carnitines were associated with lower risk of elevated blood pressure. The discussion reviews the finding in the context of what is known about metabolomics and hypertension as well as proposing possible biologic mechanisms to explain their findings.

Overall the study was well conducted with a large number of study subjects. The results add to our understanding of serum metabolome in subjects with elevated blood pressure.

One minor comment: I would have liked to know if any of the subjects in the study were on anti-hypertensive medications.

Done. Thank you so much for your appropriate comment. The number of subjects that were on anti-hypertensive medication in each group were provided in Table 1 under the name of “Antihypertensive drugs, n (%)”. We also described the number of subjects that were on anti-hypertensive medications in the result section of the revised manuscript.

Reviewer #2: Thank you for the opportunity to review the manuscript by Razi et al. In this interesting research article, link between plasma amino acids and acylcarnitines with different blood pressure categories ranging from normal BP to elevated to the different stages of hypertension were investigated. Two metabolite clusters were identified with factor analyses. The first factor consisting of various short- and medium chain acylcarnitines was associated with an increased odds of having stage 2 hypertension, while a second factor consisting of two amino acids, namely glycine and serine was associated with a lower odds of have stage 2 hypertension. I do have the following comments and suggestions:

• Overall, the manuscript would benefit greatly from professional language and technical editing.

Done. Thanks for your appropriate comment. We revised the manuscript in the sight of professional language and technical editing.

• Abstract: Line 27: What is meant with different stages of blood pressure. This should be phrased so that it is clear that you divided your groups based on different blood pressure cut points to classify normal, high and hypertensive stages.

Done, Thanks for your accurate comment. We rephrased this sentence in the revised manuscript and it is now clear that we divided the study population into 4 groups of normal blood pressure, elevated blood pressure, stage 1 and stage 2 hypertension. 

• Abstract: Which criteria did you use for blood pressure classification

Done. Thanks for your appropriate comment. We declared in the abstract section of the revised manuscript that we used ACC/AHA hypertension guidelines to classified our study population.

• Abstract: Line 36-37: The study design and statistical approach does not allow you to determine cause and effect. Therefore, you should rephrase: “to estimate the effect of metabolites on the risk of hypertension.”

Done. You are absolutely correct. We rephrased this sentence in the revised manuscript. We used the association between two variables instead of causality. 

• Abstract: Perhaps indicate the confounders you adjusted for.

Done. Thanks for your comment. We indicated the confounders we adjusted for in the abstract section of the revised manuscript.

• Abstract: Be more specific regarding the indexes (or factors) you identified. It is stated in a much clearer fashion in the conclusion of the article.

Done. You are right. Thanks for your appropriate comment We specified the factors in the result of abstract section in the revised manuscript.

• Introduction: Line 53: “which shows a significant growth trend.” Can perhaps be rephrased to indicate that there was an upward trajectory.”

Done. Thank you for your comment. We rephrased it in the revised manuscript.

• Introduction: Line 57: “….it might be ignored.” It is not clear what is meant here? What might be ignored?

Done. Thanks for your appropriate comment. We rephrased it in the revised manuscript. We meant that HTN may not be diagnosed due to variation in blood pressure, observer bias, and using nonstandard devices.

• Introduction: Line 60-62: The mention of biomarkers not related to metabolomics per se is not applicable here. Perhaps you should rather refer to what was found in previous metabolomics papers?

Done. You are absolutely right. We omitted the unrelated biomarkers in the revised introduction and instead, we referred to the metabolomics papers.

• Introduction: Line 61-62: “….no parameters have been confirmed for diagnosing and predicting hypertension.” This statement is simply not true as there are many factors previously reported on (including biomarkers) that can be used for the prediction of hypertension. Perhaps I misunderstood you. Please clarify.

Done. Thanks for your appropriate comment. You are right. We deleted this statement in the revised manuscript.

• Introduction: Line 66: What is meant with metabolomics is a practical method? It requires expensive equipment and trained staff to perform these analyses?

Done. Thanks for your precise comment. We corrected it in the revised manuscript.

• Introduction: Line 68: The reference to organic acids is quite broad as there are many organic acids that form part of multiple metabolic pathways?

Done. Thanks for your suitable comment. We revised it in the introduction section of the revised manuscript.

• Methods: Line 83: Please specify the device used to measure BP

Done. Thanks for your appropriate comment. We specified the device used to measure BP in the method section. We used Lumiscope professional aneroid sphygmomanometer with adult cuff and stethoscope to measure BP.

• Methods: For both the biochemical measurements and the metabolomics analysis the coefficient of variation should be reported. 

Done. Thank you for your appropriate comment. We reported the coefficient of variation in the revised manuscript.

• Methods: What type of QCs did you incorporate in your metabolomics analyses. Was it done in batches? How did you correct for batch effects? 

Done. Thanks for your comment. All materials were bought at the same time (first of the study), so batch difference did not affect the results. To ensure the reliability of results, quality control material were analyzed together with sample in each run which is briefly explained in revision.

• Methods: Line 101: Reference is made to internal standards. What was used? What concentrations? 

Done. Thank you for the useful comment. Internal standards were used to compensate variation happened during sample preparation, extraction and chemical derivatization and also to decrease variability in signal intensity. The concentration for each analyte was different, briefly 3-22 µmol/L for aminoacids and 0.03-0.7 µmol/L for acylcarnitines. 

• Methods: What is the level of certainty of the identification of your metabolites? Did you compare it with commercially available standards? How did you quantify the levels of your metabolites?

Done. Thank you for the comment. We used commercial standards /calibrators and prepared different dilutions to make calibration curve based on response of STD/ IS for each analyte. Using the calibration curve, the value of each analyte was quantified in each sample. To ensure the reliability of results, quality control material were analyzed together with sample in each run. To clarify the process of calibration and quality control, it is briefly explained in the revision.

• Methods: Line 121: Adjustments were made for the lipid profile? Did you include all measures as part of the lipid profile? Or only HDL? Be specific.

Done. Thanks for your excellent comment. We adjusted for age, sex, BMI, lipid profile (cholesterol, triglyceride, HDL-C), FPG, HbA1c, oral glucose-lowering drugs use, and statin use in the model 2 and we clearly stated these covariates in the method section of the revised manuscript.

• Methods: Line 121: How did you decide on FPG to adjust for instead of HbA1c?

Done. Thanks for your comment. We adjusted our model 2 for age, sex, BMI, lipid profile (cholesterol, triglyceride, HDL-C), FPG, HbA1c, oral glucose-lowering drugs use, and statin use. We did this adjustment in all of our analysis including model 2 in logistic regression analysis (S2 Table) and model 2 in our factor analysis (Fig 2). We also revised the main text, S2 Table, and Fig 2 according to the new results from model 2 adjustment.

• Results: Line 127-128: Instead of indication that each group consisted of almost half females, rather indicate that there was an equal sex distribution.

Done. Thanks for your comment. We indicated in the result section of the revised manuscript that there was an equal sex distribution instead of each group consisted of almost half females.

• Results: Line 131: I am not sure where the (p≤0.026) comes from as all the p-trend values indicated in Table 1 (with the exception of statin use) are 0.001? Did you also compare the groups in post-hoc analyses? If you did, it should be indicated in Table 1.

Done. Thank you very much for your comment. We correctly stated the p-value in the revised manuscript. We provided the results of post-hoc analyses in the Table S1. The (p≤0.026) referred to the pairwise comparison of stage 2 HTN with control and stage 1 HTN with control for the following parameters: BMI, waist circumference, FPG, HbA1C, TG, total cholesterol, and Non-HDL-C. We clearly provided all pairwise comparisons between groups in the Table S1.

• Did you adjust for statin use or the use of other medication? Perhaps that should be included as sensitivity analyses?

Done. Thanks for your appropriate comment. In the revised manuscript, we adjusted our model 2 for age, sex, BMI, lipid profile (cholesterol, triglyceride, HDL-C), FPG, HbA1c, oral glucose-lowering drugs use, and statin use. We revised the main text, figure 2 and S2 Table based on the new adjusted model 2.

• What was the rationale for inter-correlating the metabolites (Fig1). As this was not part of your original aim? 

Done. Thanks for your question. As you have well mentioned, it is not an aim of this study but it is a preliminary fundamental step for further data analysis which the whole analysis is based on. The existence of enter-correlation supported the need of PCA, as it is mentioned in the method part.

• Table 2: It is not clear what exactly you are reporting. Did you compare all groups? Or did you compare all groups with the control group? I think it may be worthwhile to do the latter if you did not. Also, to just indicate that there are differences without indicating if it was higher/lower does not say much?

Done. Thanks for your promising comment. In this table, we compared all group. But now, we provided the pairwise comparisons of metabolite concentration in different groups with control group in the last columns of the table 2. In the main text, we also indicate the higher/lower level of metabolite profile in pairwise comparison of different groups with control group.

• Discussion: Line 205 & 246: I think the use of the work “upregulated” is not appropriate here. Rather say that the abundance of acylcarnitines were higher. Normally you would use the term upregulated for an enzyme and in this case if acylcarnitines are accumulated it would suggest that the beta-oxidation enzymes are not functioning optimally?

Done. Thanks for your accurate comment. We corrected this statement in the revised manuscript.

• Discussion: Line 205-207: It is indicated that beta-oxidation may not be functionally optimally. Can you speculate as to why this is the case? 

Done. Thanks for your comment. It can be caused when the fatty acids uptake surpasses the rates of β-oxidation or in the cases of incomplete fatty acids β-oxidation which leads to intracellular lipids accumulation (reference 15).

• Discussion: Line 254: It is indicated that plasma is the best sample choice. It should be considered that urine can also be used and that urine is a lot less invasive to collect. 

Done. Thanks for your appropriate comment, this part was deleted.

• Conclusion: Line 261: “….found the unique metabolite profile of plasma” This phrase in unclear. Also the phrase “….different stages of blood pressure” is not clear and should be clarified.

Done. Thanks for your appropriate comment. We clearly rephrased this sentence in the revised manuscript.

---

## [Decision Letter · Decision Letter 1]

4 Oct 2022

PONE-D-21-34134R1Association of plasma acylcarnitines and amino acids profile with hypertension: a nationwide metabolomics studyPLOS ONE

Dear Dr. Razi,

Thank you for submitting your manuscript to PLOS ONE. After careful consideration, we feel that it has merit but does not fully meet PLOS ONE’s publication criteria as it currently stands. Therefore, we invite you to submit a revised version of the manuscript that addresses the points raised during the review process.

After the second revision, the 2 reviewers came to different conclusions.  I side with reviewer 2 as there are still several issues related to the overall grammar of the manuscript as well as the details in the method section.  Please review all comments and provide a detailed rebuttal.  ==============================

We look forward to receiving your revised manuscript.

Kind regards,

Timothy J Garrett, PhD

Academic Editor

PLOS ONE

Reviewers' comments:

Reviewer's Responses to Questions

**Comments to the Author**

1. If the authors have adequately addressed your comments raised in a previous round of review and you feel that this manuscript is now acceptable for publication, you may indicate that here to bypass the “Comments to the Author” section, enter your conflict of interest statement in the “Confidential to Editor” section, and submit your "Accept" recommendation.

Reviewer #1: All comments have been addressed

Reviewer #2: (No Response)

2. Is the manuscript technically sound, and do the data support the conclusions?

Reviewer #1: Yes

Reviewer #2: Partly

3. Has the statistical analysis been performed appropriately and rigorously? 

Reviewer #1: Yes

Reviewer #2: No

4. Have the authors made all data underlying the findings in their manuscript fully available?

Reviewer #1: Yes

Reviewer #2: Yes

5. Is the manuscript presented in an intelligible fashion and written in standard English?

Reviewer #1: Yes

Reviewer #2: No

6. Review Comments to the Author

Reviewer #1: The authors have addressed my comments. The quality of the writing could be improved but overall I can easily understand the study including results and discussion.

Reviewer #2: Thank you for the opportunity to review the revised manuscript by Razi et al. Thank you for addressing the comments, however there is still several issues with this manuscript.

• Despite claiming that the manuscript was revised to incorporate professional language and technical editing. There are still several language errors.

• The term “Non-HDL-C” is rather vague? What exactly is meant with Non-HDL-C?

• Based on the comments of Reviewer 1 you now added Antihypertensive drug use and in the discussion of the article you elaborate on how certain anti-hypertensive drugs can modify acylcarnitine levels. This brings about a lot of questions such as what type of drugs did your population use? Should you not consider adjusting for this in the statistical analyses? Or do a sensitivity analysis where those using antihypertensive drugs are excluded?

• Thank you for clarifying the adjustment for the lipid profile. However, adjustment for cholesterol, triglycerides, and HDL-C in one model may lead to multicollinearity as these are naturally inter-correlated. This may inflate the strength of your models. My suggestions would be to select the lipid with the most relevance to the dependent and main independent variables. This can be based on the literature or by doing bi-variate regression analysis between the dependent and main independent variables with the various lipids. Or multicollinearity should be tested and indicated in the manuscript.

• Similarly, the inclusion of both FGP and HbA1c may also lead to multicollinearity.

• Just check the following phrase in the discussion (line 275-276): “Our crude and adjusted models indicted that glycine had a protective effects against the stage 2 HTN incident.” Not only does this sentence not make sense, but the phrase “protective effects” cannot be used in the context of a cross-sectional analyses.

• Please supply the evidence for the following statement: “This study had several significant strengths including its large-scale population-based design, investigation of a large number of acylcarnitine and amino acids profile, evaluation of metabolite in plasma which is the best sample for this purpose…..”

7. PLOS authors have the option to publish the peer review history of their article (what does this mean?). If published, this will include your full peer review and any attached files.

Reviewer #1: **Yes: **Jean-Paul Kovalik

Reviewer #2: **Yes: **Catharina MC Mels

---

## [Author Response · Author response to Decision Letter 1]

17 Nov 2022

Response to Reviewers

Reviewer #1: The authors have addressed my comments. The quality of the writing could be improved but overall I can easily understand the study including results and discussion.

Done. Thank you. We improved the quality of the writing in the revised manuscript.

Reviewer #2: Thank you for the opportunity to review the revised manuscript by Razi et al. Thank you for addressing the comments, however there is still several issues with this manuscript.

• Despite claiming that the manuscript was revised to incorporate professional language and technical editing. There are still several language errors.

Done. Thank you for your comment. The manuscript was revised to incorporate professional language and technical editing.

• The term “Non-HDL-C” is rather vague? What exactly is meant with Non-HDL-C?

Thank you for your comment. For clarity, we replaced non-HDL-C by non-HDL cholesterol in the revised manuscript and described how non-HDL cholesterol was calculated. Non-HDL cholesterol was simply calculated by subtraction of high-density lipoprotein (HDL, or "good") cholesterol number from the total cholesterol number. So non-HDL cholesterol contains all the "bad" types of cholesterol.

• Based on the comments of Reviewer 1 you now added Antihypertensive drug use and in the discussion of the article you elaborate on how certain anti-hypertensive drugs can modify acylcarnitine levels. This brings about a lot of questions such as what type of drugs did your population use? Should you not consider adjusting for this in the statistical analyses? Or do a sensitivity analysis where those using antihypertensive drugs are excluded?

Done. Thanks for your appropriate comment. You are absolutely right. We adjusted for using antihypertension drugs in the Model 2. When we included antihypertensive drugs intake in our full adjustment model, three metabolites, including C8:1, C10, and C10:1, lost their predictive value for stage 2 HTN. This indicated that the association of these metabolites with stage 2 HTN was dependent of antihypertensive drugs intake. Finally, we had three models including crude model, model 1 that adjusted for age, sex and BMI, and model 2 that adjusted for age, sex, BMI, lipid profile (total cholesterol, triglyceride, HDL cholesterol), FPG, oral glucose-lowering drugs use, statins use, and antihypertensive drugs use. We also added it to our limitations that the type of antihypertension drugs that the participants consumed was unclear.

• Thank you for clarifying the adjustment for the lipid profile. However, adjustment for cholesterol, triglycerides, and HDL-C in one model may lead to multicollinearity as these are naturally inter-correlated. This may inflate the strength of your models. My suggestions would be to select the lipid with the most relevance to the dependent and main independent variables. This can be based on the literature or by doing bi-variate regression analysis between the dependent and main independent variables with the various lipids. Or multicollinearity should be tested and indicated in the manuscript.

Done. Thank you for your appropriate comment. We tested the multicollinearity between total cholesterol, triglycerides, and HDL cholesterol and we provided the results of the multicollinearity tests in the revised manuscript. As shown, the magnitudes of correlation coefficient between total cholesterol, triglycerides, and HDL cholesterol were lower than 0.5; so, we adjusted for all lipid profile (total cholesterol, triglycerides, and HDL cholesterol) in our model due to low multicollinearity and also, because all these lipid profile (total cholesterol, triglycerides, and HDL cholesterol) independently were associated with HTN (1, 2) and metabolite concentrations (3).

• Similarly, the inclusion of both FGP and HbA1c may also lead to multicollinearity.

Done. Thank you for your appropriate comment. There was high linear relationship between FGP and HbA1c (Pearson correlation coefficient=0.805) and so there was multicollinearity between them. These variables both represent the glucose concentration. The previous evidence in the literature showed that impaired FPG only showed a 2.13-fold increased risk for HTN while Impaired HbA1C only was not found to be independently associated with HTN (4). In addition, the study by Alqudah A et al. showed that FPG had stronger correlation with various metabolites concentration than HbA1c (5). So, we only adjusted for FPG in the model 2.

• Just check the following phrase in the discussion (line 275-276): “Our crude and adjusted models indicted that glycine had a protective effects against the stage 2 HTN incident.” Not only does this sentence not make sense, but the phrase “protective effects” cannot be used in the context of a cross-sectional analyses.

Done. Thank you for your appropriate comment. You are absolutely correct. We cannot conclude a causal effect from cross-sectional study. So, we corrected this sentence in the revised manuscript, and now the revised sentence (“Our crude and adjusted models indicated that glycine was negatively associated with the stage 2 HTN.”) showed a relationship effect.

• Please supply the evidence for the following statement: “This study had several significant strengths including its large-scale population-based design, investigation of a large number of acylcarnitine and amino acids profile, evaluation of metabolite in plasma which is the best sample for this purpose…..”

Done. Thank you for your comment. We revised this statement in the revised manuscript. This study investigated the metabolomics analysis of 50 metabolites (acylcarnitines and amino acids) on 1200 participants, and to our knowledge, no metabolomics analysis study has been conducted in Iran with such a large sample size and metabolite count as this study. Besides, there have been few studies in the world with sample size of over 1200 participants that investigated the metabolomics level of over 50 acylcarnitines/amino acids. 

1. Bønaa K, Thelle D. Association between blood pressure and serum lipids in a population. The Tromsø Study. Circulation. 1991;83(4):1305-14.

2. Tohidi M, Hatami M, Hadaegh F, Azizi F. Triglycerides and triglycerides to high-density lipoprotein cholesterol ratio are strong predictors of incident hypertension in Middle Eastern women. Journal of human hypertension. 2012;26(9):525-32.

3. Yang P, Hu W, Fu Z, Sun L, Zhou Y, Gong Y, et al. The positive association of branched-chain amino acids and metabolic dyslipidemia in Chinese Han population. Lipids in health and disease. 2016;15(1):1-8.

4. Geva M, Shlomai G, Berkovich A, Maor E, Leibowitz A, Tenenbaum A, et al. The association between fasting plasma glucose and glycated hemoglobin in the prediabetes range and future development of hypertension. Cardiovascular diabetology. 2019;18(1):1-9.

5. Alqudah A, Wedyan M, Qnais E, Jawarneh H, McClements L. Plasma amino acids metabolomics' important in glucose management in type 2 diabetes. Frontiers in pharmacology. 2021;12.

---

## [Decision Letter · Decision Letter 2]

15 Dec 2022

Association of plasma acylcarnitines and amino acids with hypertension: a nationwide metabolomics study

PONE-D-21-34134R2

Dear Dr. Razi,

We’re pleased to inform you that your manuscript has been judged scientifically suitable for publication and will be formally accepted for publication once it meets all outstanding technical requirements.

Kind regards,

Timothy J Garrett, PhD

Academic Editor

PLOS ONE

Additional Editor Comments (optional):

Reviewers' comments:

Reviewer's Responses to Questions

**Comments to the Author**

1. If the authors have adequately addressed your comments raised in a previous round of review and you feel that this manuscript is now acceptable for publication, you may indicate that here to bypass the “Comments to the Author” section, enter your conflict of interest statement in the “Confidential to Editor” section, and submit your "Accept" recommendation.

Reviewer #1: All comments have been addressed

2. Is the manuscript technically sound, and do the data support the conclusions?

Reviewer #1: Yes

3. Has the statistical analysis been performed appropriately and rigorously? 

Reviewer #1: Yes

4. Have the authors made all data underlying the findings in their manuscript fully available?

Reviewer #1: Yes

5. Is the manuscript presented in an intelligible fashion and written in standard English?

Reviewer #1: Yes

6. Review Comments to the Author

Reviewer #1: No further comments from me, all of my concerns have been addressed. Apparently I must have 100 characters in this response.

7. PLOS authors have the option to publish the peer review history of their article (what does this mean?). If published, this will include your full peer review and any attached files.

Reviewer #1: **Yes: **Jean-Paul Kovalik

---

## [Editor Report · Acceptance letter]

5 Jan 2023

PONE-D-21-34134R2 

Association of plasma acylcarnitines and amino acids with hypertension: a nationwide metabolomics study 

Dear Dr. Razi:

I'm pleased to inform you that your manuscript has been deemed suitable for publication in PLOS ONE. Congratulations! Your manuscript is now with our production department. 

Kind regards, 

on behalf of

Dr. Timothy J Garrett 

Academic Editor

PLOS ONE